# Sex disparities of human neuromuscular decline in older humans

Yuxiao Guo[1,2], Eleanor J. Jones[2] ⓘ, Thomas F. Smart[2] ⓘ, Abdulmajeed Altheyab[2,3] ⓘ, Nishadi Gamage[2,4], Daniel W. Stashuk[5] ⓘ, Jessica Piasecki[6] ⓘ, Bethan E. Phillips[2] ⓘ, Philip J. Atherton[2] ⓘ and Mathew Piasecki[2] ⓘ

[1] *Institute of Sports Medicine and Health, Chengdu Sport University, Chengdu, China*
[2] *Centre of Metabolism, Ageing & Physiology (COMAP), MRC-Versus Arthritis Centre for Musculoskeletal Ageing Research, National Institute for Health Research (NIHR) Nottingham Biomedical Research Centre, School of Medicine, University of Nottingham, Nottingham, UK*
[3] *College of Applied Medical Sciences, King Saud bin Abdulaziz University for Health Science, Riyadh, Saudi Arabia*
[4] *Neurophysiology of Human Movement Group, Faculty of Health and Medical Sciences, University of Adelaide, Adelaide, Australia*
[5] *Department of Systems Design Engineering, University of Waterloo, Waterloo, Ontario, Canada*
[6] *Musculoskeletal Physiology Research Group, Sport, Health and Performance Enhancement Research Centre, Nottingham Trent University, Nottingham, UK*

The peer review history is available in the Supporting information section of this article (https://doi.org/10.1113/JP285653#support-information-section).

**Abstract** Females typically live longer than males but, paradoxically, spend a greater number of later years in poorer health. The neuromuscular system is a critical component of the progression to frailty, and motor unit (MU) characteristics differ by sex in healthy young individuals and may adapt to ageing in a sex-specific manner due to divergent hormonal profiles. The purpose of this

This article was first published as a preprint. Guo Y, Jones EJ, Smart TF, Altheyab A, Gamage N, Stashuk DW, Piasecki J, Phillips BE, Atherton PJ, Piasecki M. 2023. Sex disparities in age-related neuromuscular decline: unveiling female susceptibility from early to late elderly. bioRxiv. https://doi.org/10.1101/2023.06.13.544761

The Journal of Physiology

study was to investigate sex differences in vastus lateralis (VL) MU structure and function in early to late elderly humans. Intramuscular electromyography signals from 50 healthy older adults (M/F: 26/24) were collected from VL during standardized submaximal contractions and decomposed to quantify MU characteristics. Muscle size and neuromuscular performance were also measured. Females had higher MU firing rate (FR) than males ($P = 0.025$), with no difference in MU structure or neuromuscular junction transmission (NMJ) instability. All MU characteristics increased from low- to mid-level contractions ($P < 0.05$) without sex $\times$ level interactions. Females had smaller cross-sectional area of VL, lower strength and poorer force steadiness ($P < 0.05$). From early to late elderly, both sexes showed decreased neuromuscular function ($P < 0.05$) without sex-specific patterns. Higher VL MUFRs at normalized contraction levels previously observed in young are also apparent in old individuals, with no sex-based difference of estimates of MU structure or NMJ transmission instability. From early to late elderly, the deterioration of neuromuscular function and MU characteristics did not differ between sexes, yet function was consistently greater in males. These parallel trajectories underscore the lower initial level for older females and may offer insights into identifying critical intervention periods.

(Received 8 September 2023; accepted after revision 7 May 2024; first published online 10 June 2024)

**Corresponding author** M. Piasecki: Centre of Metabolism, Ageing & Physiology (COMAP), Academic Unit of Injury, Inflammation & Recovery Sciences, School of Medicine, Faculty of Medicine & Health Sciences, University of Nottingham, Royal Derby Hospital Centre (Room 3011), Derby, DE22 3DT, UK. Email: Mathew.piasecki@nottingham.ac.uk

**Abstract figure legend** Intramuscular electromyography applied to the vastus lateralis of older males (pink) and females (purple) during low- and mid-level voluntary contractions reveal higher motor unit firing rates in females. In both sexes the decline of neuromuscular from early to late elderly follow similar trajectories yet is consistently higher in males.

### Key points

- Females generally exhibit an extended lifespan when compared to males, yet this is accompanied by a poorer healthspan and higher rates of frailty.
- In healthy young people, motor unit firing rate (MUFR) at normalized contraction intensities is widely reported to be higher in females than in age-matched males.
- Here we show in 50 people that older females have higher MUFR than older males with little difference in other MU parameters. The trajectory of decline from early to late elderly does not differ between sexes, yet function is consistently lower in females.
- These findings highlight distinguishable sex disparities in some MU characteristics and neuromuscular function, and suggest early interventions are needed for females to prevent functional deterioration to reduce the ageing health–sex paradox.

## Introduction

The well-established sex differences of ageing indicate that females typically live longer than males yet do so in poorer health, with lower levels of physical performance and a higher frailty index score (Carmel, 2019; Gordon et al., 2017). Though ageing is generally characterized by decreased muscle mass and strength, there is a significant

**Yuxiao Guo** is currently a principal investigator in the Institute of Sports Medicine and Health in Chengdu Sport University. Before her time in China, she earned a PhD from the Centre of Metabolism, Ageing and Physiology (COMAP) within the University of Nottingham, UK. Her research focused on human motor unit characteristics and physical function in the context of sex and age. With a central focus on neuromuscular physiology, her current work extends into exploring the mechanisms that govern motoneuron excitability in humans and how this may adapt and be manipulated in older age.

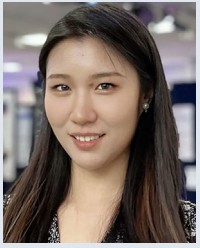

heterogeneity with respect to physical function. Yet there are limited mechanistic data on neuromuscular function in older females who are underrepresented in studies of human physiology (Cowley et al., 2021; Woitowich et al., 2020), which limits exploration of sex disparities and prevents the implementation of targeted interventions.

Among the most underexplored elements accounting for sex-specific physical performance are alterations of motor unit (MU) properties. Voluntary contraction of muscles relies upon the coordinated activation of MUs, sets of muscle fibres innervated by a single motoneuron (Heckman & Enoka, 2012), which can be recruited based on their size and further modulated via the frequency at which they discharge, referred to as MU recruitment and firing rate (FR) modulation, respectively (Enoka & Duchateau, 2017). There were ~30% fewer MUs in the quadriceps and tibialis anterior (TA) muscles of older males compared to younger males (Piasecki et al., 2016, 2018), which contributes to the loss of functionality in older age. Following denervation, a muscle fibre may atrophy and eventually be lost, or it may be 'rescued' by an adjacent surviving axon and the formation of a new neuromuscular junction (NMJ). This process of MU remodelling results in aged MUs that are fewer in number but larger in size (Jones et al., 2022). These MU adaptations may alter neuromuscular recruitment strategies and/or further influence force control (Pethick & Piasecki, 2022), and may do so in a sex-specific manner.

The hormonal milieu is one of the most distinguishing features of the two sexes, with the predominant female sex hormones oestrogen and progesterone fluctuating across a normal menstrual cycle, whilst the male sex hormone testosterone remains relatively constant (Hunter, 2014; Piasecki et al., 2024). Testosterone has an anabolic impact on skeletal muscle and its precursor dehydroepiandrosterone (DHEA) was positively associated with MUFR in highly active and inactive older males (Guo, Piasecki, et al., 2022). Oestrogen elicits excitatory effects via the potentiation of glutamatergic receptors (Smith & Woolley, 2004) while progesterone increases activity of gamma-aminobutyric acid (GABA), causing inhibitory effects (del Río et al., 2018; Smith et al., 1989). Collectively, this may partly explain alterations of neuromuscular function across the menstrual cycle (Ansdell et al., 2019; Piasecki et al., 2023). Following the menopause and the cessation of the menstrual cycle, typically occurring at the age of ~51 years (Hall, 2015), oestrogen concentrations plummet and generate a severe change in the hormonal milieu that is not apparent in males. The exclusive manifestation of this pronounced change in females may possibly generate an impaired equilibrium between neuronal excitation and inhibition in advanced age. Occurring concurrently with an increased risk of osteoporosis (Sözen et al., 2017) and heightened joint stiffness (Chidi-Ogbolu & Baar, 2019; Leblanc et al.,

2017), physical function and mobility decrements in females may exceed that of males.

We have previously reported a significant decline in MUFR in TA from middle to older highly active females, which was not observed in highly active males (Piasecki, Inns, et al., 2021). Several recent studies of healthy young people report a higher MUFR in females in vastus lateralis (VL) (Guo, Jones, et al., 2022), TA (Inglis & Gabriel, 2020; Taylor et al., 2022) and first dorsal interosseous (Nishikawa et al., 2024). Females also recruit MUs later than males during graded contractions of the lower leg muscles (Jenz et al., 2023). These consistent findings across sexes in young may highlight a potential influence of hormones, which may be amplified in older age. Furthermore, age comparisons are frequently conducted through cross-sectional studies comparing young and old groups, wherein the older group typically encompasses an age range where physiological deterioration of MU function may be anticipated (Hirono et al., 2024)

The purpose of the present study was to explore the influences of sex and ageing on VL MU characteristics and neuromuscular function of early to late elderly males and females. It was hypothesized that females would have lower functional performance than males, and markers of more extensive MU remodelling and physical deterioration than males.

## Methods

### Ethical approval

This study was approved by the University of Nottingham Faculty of Medicine and Health Sciences Research Ethics Committee (90-0820, 407-1910, 390-1121) and was conducted between 2019 and 2024 in accordance with the *Declaration of Helsinki*, except for registration in a database.

Twenty-six healthy males and twenty-four healthy females between the ages of 60 and 83 years were recruited from the local community via advertisement. All recruited participants were recreationally active on a day-to-day basis. Prior to enrolment, all participants completed a comprehensive clinical screening examination based at the School of Medicine, Royal Derby Hospital Centre and subsequently provided written informed consent. The screening procedure allowed exclusion of participants with musculoskeletal abnormalities, acute cerebrovascular or cardiovascular diseases, uncontrolled hypertension, or metabolic disease.

### Anthropometry

Prior to testing, calibrated scales and a stadiometer were used to assess the body mass and height of each participant for calculation of body mass index. Cross-sectional area

(CSA) of the VL was measured using an ultrasound probe (LA532 probe B-mode, frequency range 26−35 Hz and MyLab$^{TM}$50 scanner, Esaote, Genoa, Italy). To quantify VL CSA, images were captured from the aponeurosis borders of the VL of the right leg in a medial–lateral fashion using panoramic imaging (VPAN) and further analysed with ImageJ software (National Institutes of Health, Bethesda, MD, USA) by tracing around the VL following the contour of the aponeurosis. CSA was determined by analysing each image three times and taking the average of three images.

## Balance

A Footscan plate (Footscan, 200 Hz, RScan International, Olen, Belgium) was used to assess the postural sway during right-leg standing with eyes open. Participants were asked to step on the platform and visually focus on a target point in front of them for the duration of the test (10 s). A 5 s countdown was given before instruction to lift the left leg. Travelled distance and ellipse area of the centre of pressure (COP) were recorded. Each participant was allowed three attempts, and the best value was reported. The total distance travelled by the COP during the test time was determined as COP travelled distance (mm). COP ellipse area was calculated based on the data in the ellipse set for 95% of the total area covered by the COP trajectory in both anterior–posterior and medial–lateral directions. Smaller area and/or travelled distance indicated a better postural balance. Balance data are available for 24 males and 21 females.

## Grip strength

A handgrip dynamometer (Grip-D, Takei, Japan) was used to test the grip strength of the dominant hand. When measuring grip strength, the dynamometer was held in a standing position with the pointer facing outward, and the width of the grip was adjusted so that the index finger's interphalangeal joint was bent 90°. The arms were lowered naturally, the feet were hip width apart and the dynamometer was grasped with maximum force without touching the body or clothing. The maximal value among three measurements was accepted as the grip strength. Grip strength data are available for 24 males and 22 females.

## Timed up and go (TUG)

The TUG test was used to evaluate dynamic balance and functional mobility. The TUG test involved getting up from a standard chair without armrests, walking for 3 m, turning around an obstacle and returning to the original sitting position as quickly as possible without running. The test was initiated with a verbal 'go' instruction from the researcher, and the time taken to complete the test was recorded. After a familiarization attempt, the participant was asked to have a real go after 1 min of rest.

## Knee extensor torque

Participants were placed in a custom-built chair which was altered to ensure the hips and knees were flexed at ∼90°. The right lower leg was attached to a force dynamometer with a non-compliant strap (purpose-built calibrated strain gauge, RS125 Components Ltd, Corby, UK) above the medial malleolus. The hips and pelvis were also securely held by a seat belt to reduce movement of the hips and upper trunk during contractions. The distance between the centre of the force strap and the lateral femoral condyle was measured to determine the external knee joint moment arm. With real-time visual feedback and verbal encouragement, participants were instructed to perform each trial with maximal effort as hard and fast as they could following a standardized warm-up of sub-maximal contractions. During the trial, participants were not allowed to hold onto the side of the chair and were asked to cross their arms across the chest. It was further repeated two to three times with 60 s rest intervals between each one. If there was a difference of less than 5% between the two last attempts, the highest value was accepted as the maximal voluntary contraction (MVC). Torque was then determined by multiplying the selected MVC by the lever arm. To quantify force steadiness, a single target line at a normalized contraction intensity was displayed on a screen and the participant was instructed to follow as closely as possible. The coefficient of variation of the force (CoV) was calculated as CoV = (SD/mean) × 100. When calculating CoV, in an attempt to reduce corrective actions, the first two passes of the target (<1 s) were excluded from the analysis.

## Intramuscular electromyography (iEMG)

Prior to intramuscular needle electrode insertion, a familiarization trial was performed in which the participant had an attempt to contract to each sub-maximal target line at 10% and 25% of MVC observed on a monitor. Following this, a 25 mm or 40 mm (dependent on visible subcutaneous thickness) disposable concentric needle electrode (N53153; Teca, Hawthorne, NY, USA) was inserted into the muscle belly of the VL of the right leg, to a depth of 1.5–3.5 cm depending on muscle/thigh size. A ground electrode was placed over the patella of the same leg. To ensure an adequate signal to noise ratio once the needle was positioned, participants were asked to perform several low-level voluntary contractions until spikes were visible in real-time. Participants were then asked to perform sustained voluntary isometric contrac-tions at 10% and 25% of MVC, four times each, with

20 s intervals between contractions. Each time, the needle was repositioned 180° by twisting the bevel edge and withdrawing ∼5 mm each time to sample from spatially distinct areas (Jones et al., 2021). These contraction levels were chosen as this range is generally representative of activities of daily living such as walking and climbing stairs (Tikkanen et al., 2013) and is known to be both tolerable and to provide a high MU yield (Guo, Jones, et al., 2022). iEMG signals were recorded in Spike2 (Version 9.06), sampled at 50 kHz and bandpass filtered at 10 Hz to 10 kHz (1902 amplifier; Cambridge Electronic Design Ltd, Cambridge, UK) and stored for future off-line analysis.

### iEMG analysis

Using decomposition-based quantitative electromyography (DQEMG) software, individual motor unit potentials (MUPs) were detected, and motor unit potential trains (MUPTs) were extracted from the iEMG signal recorded during the sustained force portion of the contraction, enabling the evaluation of VL electrophysiological activity during contractions. MUPs were visually inspected and markers corresponding to MUP onset and end were adjusted if required. MUFR is reported as the estimated number of MU discharges per second (Hz) within a MUPT, with all values below 5 Hz deemed to be non-physiological and removed. MUFR variability is reported as the estimate of the CoV of the inter-discharge intervals (IDIs), displayed as a percentage. MUP area was determined as the total area within the MUP duration (onset to end). A MUP's complexity is determined by the number of phases, which are defined as the number of components above or below the baseline.

A near fibre MUP (NFM) is calculated by applying a low-pass second-order differentiator to a MUP. In this way, contributions from fibres near to the needle electrode primarily contribute to an NFM and interference from distant active fibres is reduced. All NFMs (and corresponding MUPs) without distinct peaks were excluded from analyses. NFM jiggle is a measure of the shape variability of consecutive NFMs of a MUPT expressed as a percentage of the total NFM area (Piasecki, Garnés-Camarena, et al., 2021).

### Statistics

All statistical analysis was conducted using RStudio (Version 1.3.959). Descriptive data were generated for all variables. Multiple linear regression models with age considered as a covariate were used to compare the effects of age and sex on physical characteristics. Multilevel linear regression models using the *lme4* package (Version 1.1-27.1) (Bates et al., 2015) were generated to compare MU parameters between groups across two contraction intensities in order to preserve variability between and within participants simultaneously. Sampled MUs were regarded as the first level of multilevel models, and participants with their respective clustered sets of sampled MUs were regarded as the second level. The results are displayed as coefficient estimates, 95% confidence intervals and *P*-values. Standardized estimates were calculated through the package *effectsize* (Version 0.8.2) (Ben-Shachar et al., 2020) for forest plotting. Statistical significance was accepted when $P < 0.05$.

### Results

Fifty older participants were included in the study, consisting of 26 healthy older males (age range: 61−83 years) and 24 healthy older females (age range: 60−79 years). All recorded measures are shown in Table 1.

There were no significant interactions between sex and age detected in any functional parameters. When adjusting for sex, for every year increase in age, muscle CSA decreased by 0.29 cm$^2$ [95% confidence interval (CI): −0.51 to −0.06; $P = 0.013$], muscle torque decreased by 3.10 Nm (−4.62 to −1.58; $P < 0.001$) and grip strength decreased by 0.43 kg (−0.71 to −0.15; $P = 0.004$). Though not statistically significant, for every year increase in age, TUG differed by 0.05 s (−0.01 to 0.11; $P = 0.115$). Additionally, force steadiness progressively decreased significantly at 10% (beta: −0.13; 0.05 to 0.22; $P = 0.004$) but not statistically significantly at 25% MVC (0.04; −0.004 to 0.08; $P = 0.075$). There were no significant age effects in unilateral COP travelled distance (4.27; −6.77 to 15.31; $P = 0.440$) or COP ellipse area (2.70; −1.04 to 6.44; $P = 0.152$).

When adjusting for age, females had 39.3% smaller muscle CSA (−7.24; −9.75 to −4.74; $P < 0.001$), 46.3% lower knee extensor torque (−58.72; −75.90 to −41.53; $P < 0.001$), 43.1% lower grip strength (−14.89; −18.14 to −11.63; $P < 0.001$) and 10.1% longer TUG time (0.94; 0.22 to 1.66; $P = 0.012$) than males. Females showed 21.3% and 14.5% poorer force steadiness at 10% (1.70; 0.72 to 2.69; $P = 0.001$) and 25% MVC (0.63; 0.16 to 1.10; $P = 0.009$) when compared to males. However, there were no sex differences in unilateral COP travelled distance (−87.35; −211.96 to 37.25; $P = 0.165$) or COP ellipse area (−36.92; −79.39 to 5.55; $P = 0.087$). All statistical model outputs are shown in Table 2 and individual data are shown in Fig. 1.

At 10% MVC, the mean number of sampled MUPs per person was 20 in males with a mean of 5 MUPs sampled per needle position, and 22 in females with 6 MUPs sampled per needle position. At 25% MVC, the mean number of MUPs sampled per person was 30 in males with 7 MUPs sampled per needle position, and 32 in females with 8 MUPs sampled per needle position.

**Table 1. Participant characteristics**

| Measure | | Males (*n* = 26) | Females (*n* = 24) |
|---|---|---|---|
| Age, years | | 72 (6) | 70 (5) |
| Body mass index, kg/m$^2$ | | 26.28 (2.16) | 24.89 (3.74) |
| **Physical characteristics** | | | |
| CSA, cm$^2$ | | 20.74 (5.55) | 13.93 (3.01) |
| Torque, Nm | | 144.46 (41.61) | 90.14 (24.61) |
| Grip strength, kg | | 40.39 (7.08) | 26.08 (3.70) |
| CoV force − 10% MVC | | 6.14 (1.68) | 7.60 (1.99) |
| CoV force − 25% MVC | | 3.59 (0.72) | 4.15 (0.92) |
| Timed up and go, s | | 8.11 (1.20) | 8.97 (1.35) |
| COP travelled distance, mm | | 371.83 (234.91) | 279.86 (163.30) |
| COP ellipse area, mm$^2$ | | 98.96 (75.35) | 59.40 (61.83) |
| **Motor unit properties** | | | |
| MUFR, Hz | 10% MVC | 7.90 (1.32) | 8.66 (1.53) |
| | 25% MVC | 8.23 (1.27) | 9.02 (1.10) |
| MUFR variability, % | 10% MVC | 8.48 (1.00) | 9.21 (1.23) |
| | 25% MVC | 8.84 (1.43) | 9.08 (1.28) |
| MUP phases | 10% MVC | 3.96 (0.73) | 4.02 (0.46) |
| | 25% MVC | 4.24 (0.69) | 4.28 (0.63) |
| MUP duration, ms | 10% MVC | 8.14 (1.55) | 8.15 (1.59) |
| | 25% MVC | 8.81 (1.78) | 8.64 (1.50) |
| MUP area, $\mu$V.ms | 10% MVC | 765.99 (296.93) | 615.17 (225.10) |
| | 25% MVC | 1028.09 (406.82) | 852.62 (338.52) |
| NFM jiggle, % | 10% MVC | 15.79 (3.84) | 17.13 (4.81) |
| | 25% MVC | 18.37 (6.35) | 19.26 (4.24) |

Data are reported as mean (SD).
COP, centre of pressure; CoV, coefficient of variation; CSA, cross-sectional area; FR, firing rate; MU, motor unit; MVC, maximal voluntary contraction; MUP, motor unit potential; NFM, near fibre motor unit potential.

**Table 2. Summary of multiple linear regression analysis for functional parameters**

| | Age[a] | | | Sex[b] | | |
|---|---|---|---|---|---|---|
| | Beta | 95% CI | *P*-value | Beta | 95% CI | *P*-value |
| CSA, cm$^2$ | −0.29 | −0.51 to −0.06 | **0.013** | −7.24 | −9.75 to −4.74 | **<0.001** |
| Torque, Nm | −3.10 | −4.62 to −1.58 | **<0.001** | −58.72 | −75.90 to −41.53 | **<0.001** |
| Grip strength, kg | −0.43 | −0.71 to −0.15 | **0.004** | −14.89 | −18.14 to −11.63 | **<0.001** |
| Timed up and go, s | 0.05 | −0.01 to 0.11 | 0.115 | 0.94 | 0.22 to 1.66 | **0.012** |
| CoV force – 10% MVC | 0.13 | 0.05 to 0.22 | **0.004** | 1.70 | 0.72 to 2.69 | **0.001** |
| CoV force – 25% MVC | 0.04 | −0.004 to 0.08 | 0.075 | 0.63 | 0.16 to 1.10 | **0.009** |
| COP travelled distance, mm | 4.27 | −6.77 to 15.31 | 0.440 | −87.35 | −211.96 to 37.25 | 0.165 |
| COP ellipse area, mm$^2$ | 2.70 | −1.04 to 6.44 | 0.152 | −36.92 | −79.39 to 5.55 | 0.087 |

Values in bold reflect statistically significant (*P* < 0.05) results.
CI, confidence interval; COP, centre of pressure; CoV, coefficient of variation; CSA, cross-sectional area; MVC, maximal voluntary contraction.
[a] Multiple linear regression model with **age** as the dependent variable.
[b] Multiple linear regression model with **sex** as the dependent variable.

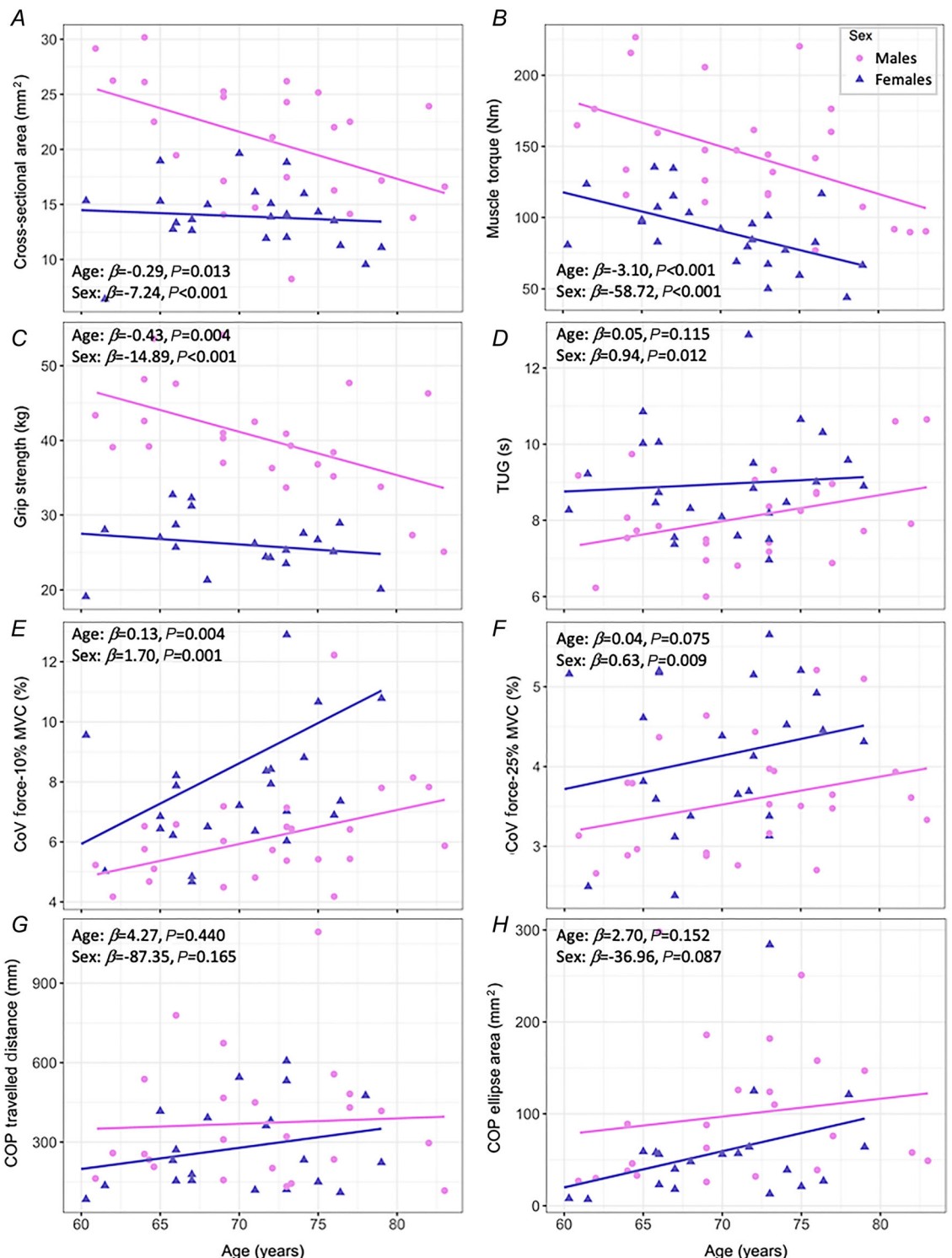

**Figure 1. Scatter points of physical parameters in older males (pink circles) and females (purple triangles)**

The estimates of the main effect of age and sex from multiple linear regression models for physical parameters with no interactions detected are reported in text boxes. For data visualization, male and female regression lines are displayed separately. COP, centre of pressure; CoV, coefficient of variation; MVC, maximal voluntary contraction; TUG, timed up and go. [Colour figure can be viewed at wileyonlinelibrary.com]

After adjusting for sex and contraction level, there was no association between age and all MU parameters (all $P > 0.05$). For data visualization, the effect of age on MU function was explored at each contraction level with an adjustment of sex (see Fig. 2).

There was no significant sex × contraction level interaction for MUFR ($P = 0.733$) but there was a main effect of sex ($P = 0.032$) with females showing a greater MUFR when compared to males. MUFR increased significantly when moving from low- to mid-level contractions in both sexes ($P = 0.001$). There was no significant sex × contraction level interaction for MUFR variability ($P = 0.086$) but there was a main effect of contraction level ($P = 0.049$) with MUFR variability increasing when moving from low- to mid-level contractions, but no significant differences detected between sexes ($P = 0.085$).

There was no significant sex × contraction level interaction for MUP phases ($P = 0.820$), and there was no statistical difference between sexes ($P = 0.647$) whereas there was a main effect of contraction level, with a greater number of MUP phases observed at 25% when compared to 10% MVC contractions in both sexes ($P < 0.001$). Similarly, there was no significant sex × contraction level interaction for MUP duration ($P = 0.233$), and there was no statistical difference between sexes ($P = 0.890$), but both sexes exhibited a longer MUP duration when moving from low- to mid-level contractions ($P < 0.001$). There was no significant sex × contraction interaction for MUP area ($P = 0.953$) and no significant main effect of sex ($P = 0.121$), but both sexes showed greater MUP area when moving from low- to mid-level contractions ($P < 0.001$).

There was no significant sex and contraction level interaction for NFM jiggle ($P = 0.962$) and no significant main effect of sex ($P = 0.440$) whereas there was a main effect of contraction level, with both sexes showing a greater NFM jiggle at 25% when compared to 10% MVC contractions ($P = 0.005$). All statistical model outputs for MU parameters are shown in Table 3 and individual data are shown in Fig. 3. Standardized regression coefficient estimates for the predicted change from 10% to 25% MVC are shown in Fig. 4.

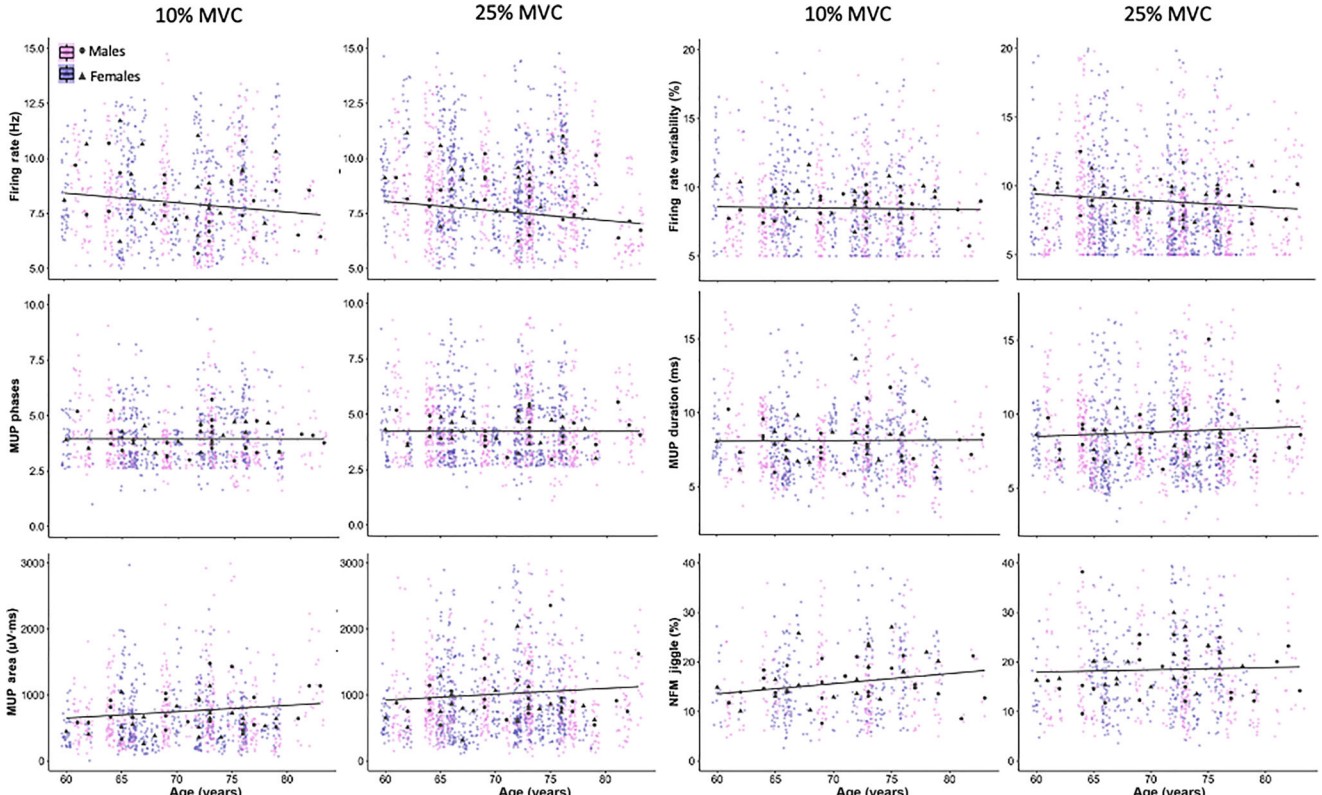

**Figure 2. Measurement of motor unit structure and function in older males (pink) and females (purple) aged 60−83 years**
All motor unit action potentials (MUPs) recorded during contractions at 10% and 25% maximal voluntary contraction (MVC) are shown as jitter points. Individual participant means are shown as scatter points (black) with males in circles and females in triangles. Grey lines indicate the beta coefficient of age from multilevel mixed effects linear regression models with an adjustment of sex. [Colour figure can be viewed at wileyonlinelibrary.com]

**Table 3. Summary of linear regression analysis for motor unit parameters**

| | Sex[a] | | | Level[b] | | |
|---|---|---|---|---|---|---|
| | Beta | 95% CI | *P*-value | Beta | 95% CI | *P*-value |
| MUFR, Hz | 0.81 | 0.09 to 1.54 | **0.032** | 0.33 | 0.14 to 0.53 | **0.001** |
| MUFR variability, % | 0.63 | −0.08 to 1.33 | 0.085 | 0.40 | 0.001 to 0.81 | **0.049** |
| MUP phases | 0.08 | −0.27 to 0.44 | 0.647 | 0.27 | 0.13 to 0.41 | **<0.001** |
| MUP duration, ms | 0.08 | −1.01 to 1.17 | 0.890 | 0.65 | 0.37 to 0.93 | **<0.001** |
| MUP area, $\mu$V.ms | −148.08 | −332.53 to 36.38 | 0.121 | 269.78 | 208.00 to 331.56 | **<0.001** |
| NF jiggle, % | 1.13 | −1.72 to 3.98 | 0.440 | 2.26 | 0.69 to 3.84 | **0.005** |

Beta value and 95% confidence interval (CI) represent the model-predicted change per unit between sexes and contraction levels. Values in bold reflect statistically significant ($P < 0.05$) results.
FR, firing rate; MU, motor unit; MUP, motor unit potential; NFM, near fibre motor unit potential.
[a] Multiple mixed-effects linear regression model with sex as the dependent variable.
[b] Multiple mixed-effects linear regression model with contraction level as the dependent variable.

## Discussion

Here we provide evidence highlighting sex disparities in VL MU characteristics in older people, as revealed through iEMG. Our study demonstrates that older females display higher MUFR at normalized contraction intensities compared to older males, without any sex-based difference observed in estimates of MU structure or NMJ transmission instability. When assessing alterations of active MUs from low- to mid-level contractions, both sexes exhibit increased MUFR, as well as estimates of MU structure and NMJ transmission instability. Additionally, the current study demonstrates an age-related decrease in physical performance in early to late elderly humans in both males and females, yet with distinct sex differences. Regardless of the progressive decrease, females had a smaller muscle size, lower knee extensor and grip strength, longer TUG time and poorer force steadiness than males. These data underscore the connection between diminished functionality in older females and a lower baseline observed during early stages of ageing, potentially serving as a valuable reference for identifying inflection points along the path towards frailty.

Females exhibited significantly higher MUFR than males at normalized contraction levels, and this difference across sex was over twofold larger than the difference from low- to mid-level contractions (beta: sex 0.81 *vs.* level 0.33). This higher MUFR in females is consistent with numerous studies of young previously observed in both upper and lower extremities (Guo, Jones, et al., 2022; Harwood et al., 2014; Inglis & Gabriel, 2021; Nishikawa et al., 2024; Peng et al., 2018; Taylor et al., 2022). We previously reported in young VL a higher MUFR in females, with a smaller MUP and similar MU number estimates. Here we suggested that females compensate for smaller MUs by discharging at higher rates (Guo, Jones, et al.

2022). However, in the current study of older people, there was no significant difference in MUP area between sexes, which may be a result of MU remodelling (i.e. expansion) occurring in both sexes (Jones et al., 2022). Therefore, the same argument of compensating for smaller MUs lacks conviction in old people.

Myosteatosis (fatty infiltration of muscle) increases in older age and is associated with impaired neuromuscular function (Aoi et al., 2020). Females typically exhibit higher levels of myosteatosis compared to males, irrespective of variations in body mass index or overall body fat levels (Goodpaster et al., 2001; Ryan et al., 2011; Therkelsen et al., 2013; Xiao et al., 2018). More specifically, as intramuscular adipose tissue increases, even when adjusted for muscle CSA, muscle quality declines (Biltz et al., 2020). Therefore, the elevated MUFR observed in older females may represent a compensatory mechanism for lower muscle quality.

Differences in MUFR may stem from the organization of central and peripheral inputs into the motoneuron pool and/or differences in the intrinsic characteristics of MUs. Persistent inward currents (PICs) play a critical role in initiating sustained firing of motoneurons under conditions of descending monoaminergic drive (Heckman et al., 2008). Estimates of the PIC contribution to MU firing made via the deltaF technique (Gorassini et al., 2002) are consistently lower in old males compared to young (Guo et al., 2024; Hassan et al., 2021; Orssatto et al., 2022, 2023) and are greater in young females compared to young males (Jenz et al., 2023). Given the sex differences of MUFR widely observed in young are also apparent in old, it is probable that estimates of the PIC contribution to MU firing also maintain a sex difference in older age. Although the deltaF technique was not used here, these data suggest minimal effects of the menopause on MUFR at these low- and mid-level contraction intensities.

This may not be true of all contraction levels and/or muscles; in lower limbs, specifically the vastus medialis, TA and VL, females exhibited higher MUFR during submaximal contractions, ranging from 10% to 75% of MVC (Guo, Jones, et al., 2022; Inglis & Gabriel, 2020; Kowalski & Anita D., 2020; Taylor et al., 2022). Conversely, at 100% MVC, females showed lower MUFR than males. Notably, when strength and physical activity were balanced between the sexes, females consistently maintained a higher MUFR than males (Inglis & Gabriel,

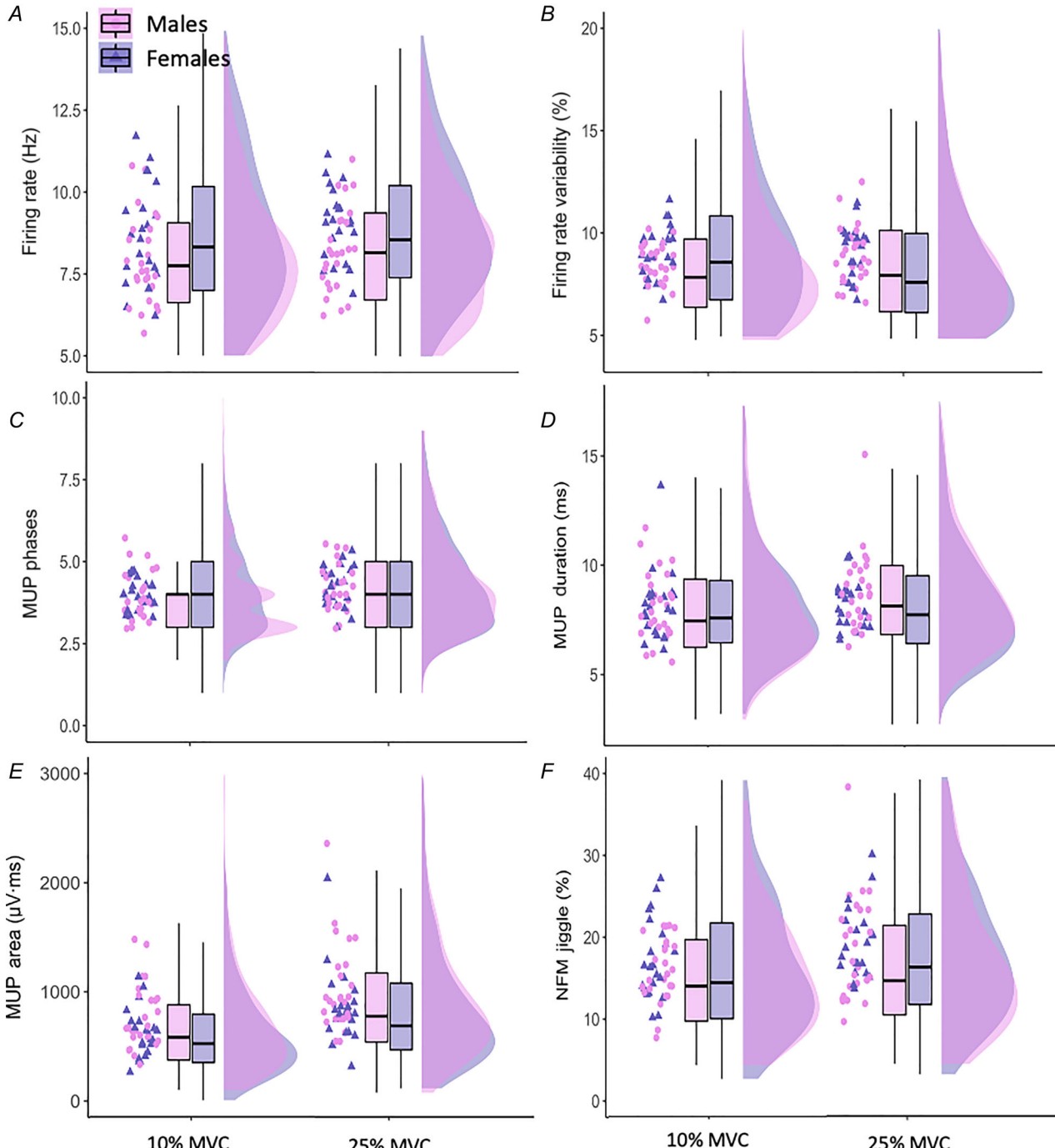

**Figure 3. Motor unit (MU) properties in males (pink) and females (purple) at 10% and 25% maximal voluntary contraction (MVC)**
Individual participant means are shown in the left column within each plot with males in circles and females in triangles. Box plots illustrate the first and third quartiles, and the median of all MUs. Distributions of all MUs are also shown in each density plot. MUP, motor unit potential; NFM, near fibre motor unit potential. [Colour figure can be viewed at wileyonlinelibrary.com]

2020). With force increases, we found that recruitment strategies for early and late elderly males and females did not differ, also reported for young cohorts (Guo, Jones, et al., 2022). Put simply, when moving from a low- to a mid-level contraction, the change in MU rate coding and recruitment did not differ between sexes, as indicated by higher MUFR and larger MUP area in males and females.

Although MUFR variability showed no statistical significance across sex ($P = 0.085$), the sex-specific difference when moving from low- to mid-level contractions is notable (see Fig. 4). This remained similar in females but increased in males, a pattern not observed in young VL (Guo, Jones, et al., 2022). In the TA of young individuals, however, females had higher MUFR variability alongside poorer force steadiness (Inglis & Gabriel, 2021), again highlighting possible muscle-specific effects of ageing.

Force fluctuations are notably more pronounced in the elderly compared to the young (Carville et al., 2007; Enoka et al., 2003). In this study, we demonstrate a significant reduction in force steadiness during low-level submaximal sustained contractions from early to late elderly. This age-related loss is probably multifactorial and includes the altered descending motor commands, altered MU discharge properties (Castronovo et al., 2018; Farina & Negro, 2015), as well as the loss and subsequent remodelling of MUs (Challis, 2006; Jones et al., 2022). Age-related changes in mechanical properties of joints also play a crucial role in regulating force and control ability. Greater muscle tone and reduced elasticity in upper limbs (Lee et al., 2022) as well as heightened co-contraction of agonist and antagonist muscles in lower

limbs (Hortobágyi & DeVita, 2006; Krishnan et al., 2011) have been widely reported with age. These changes lead to elevated joint contact forces (Hodge et al., 1986) and increased impact loads (Lafortune et al., 1996). Moreover, a decline in collagen content with age may contribute to increased tendon and ligament laxity, potentially exacerbating muscle co-contraction patterns (Rudolph et al., 2007).

Both sexes exhibited considerable improvements in force steadiness during transitions to mid-level contractions, as is to be expected as mean force values increase disproportionately to the associated SD. This may also be a result of decreased motor noise levels such as decreased fluctuations in common oscillatory synaptic inputs (Hamilton et al., 2004).

After adjusting for age, older females displayed noticeably poorer force steadiness in comparison to older males. This discrepancy may be due to the previously outlined variations in motor noise, given that females exhibit lower absolute strength. Beyond neural considerations, this sex disparity may also be explained by biomechanical variations between sexes, where post-menopausal females with reduced oestrogen levels tend to exhibit reduced tendon elasticity and flexibility, leading to a greater degree of joint instability (Chidi-Ogbolu & Baar, 2019; Leblanc et al., 2017).

Numerous studies have highlighted the age-related reduction of muscle size and strength (Piasecki et al., 2016; Wilkinson et al., 2018) and the quadriceps appear to be particularly susceptible when compared to other lower extremity muscles such as hamstrings and TA (Maden-Wilkinson et al., 2013). Interestingly, the ∼55%

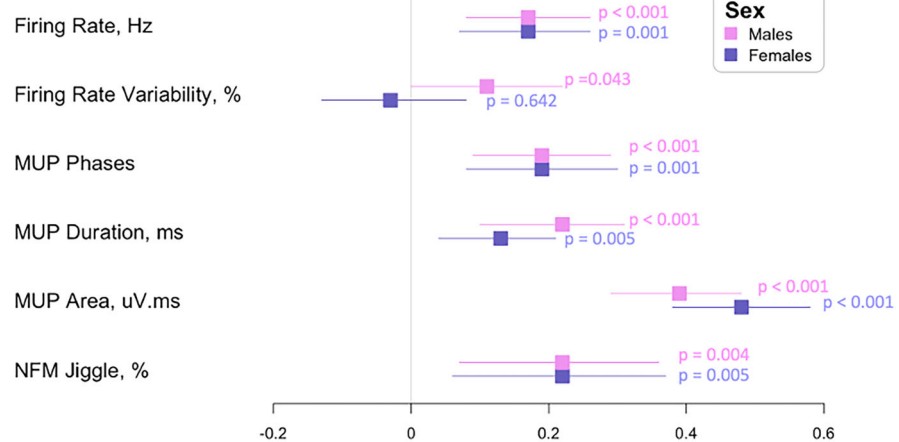

**Figure 4. Forest plot of the standardized regression coefficient estimate for associations between motor unit characteristics and contraction levels in older males (pink) and females (purple)**
Beta and 95% confidence intervals represent the standardized model-predicted change per unit from 10% to 25% maximal voluntary contraction (MVC). All statistical analysis was based on multilevel mixed effects linear regression models. Standardized values for each parameter make the comparisons between older males and females justifiable. MUP, motot unit potential; NFM, near fibre motor unit potential. [Colour figure can be viewed at wileyonlinelibrary.com]

lower muscle size and ∼48% lower strength in older females reported here is greater than the sex-based differences we have previously reported when comparing healthy young people (∼39% and ∼37%, respectively) (Guo, Jones, et al., 2022), and suggests the sex-based disparities are augmented in older age, to the detriment of females.

Grip strength has been identified as a robust predictor of frailty and mortality among older individuals (García-Hermoso et al., 2018) and the current data show a significant decline from early and late elderly. This is also in alignment with large-scale investigations (Andersen-Ranberg et al., 2009) that identified lower grip strength in females when compared to males. Even though age did not seem to have a statistically significant effect, sex-based differences in TUG scores were observed with females showing higher scores than males, indicating that females in their early and late older age have poorer physical performance.

## Strengths and limitations

This is the first study using iEMG techniques to explore individual VL MUs of early to late older males and females, enabling the neural characterization of a large muscle highly susceptible to age-related loss. This direct sex comparison contributes to the lack of data on older females and reveals that some of the functional decrements exceed those which we have previously reported in healthy young individuals. However, although iEMG enables MUP sampling from deeper muscles, it is limited to low- and mid-level contractions and these findings cannot be extrapolated to higher threshold MUs. Secondly, we did not measure the circulating levels of sex hormones nor other circulating biomarkers related to neural plasticity. Thirdly, the right leg was uniformly assessed across all participants and, as such, cannot infer possible bilateral differences which may contribute to functional impairment.

## Conclusions

Older females display notably higher MUFR at normalized contraction intensities compared to older males, without any sex-based difference observed in estimates of MU structure or NMJ transmission instability. In both sexes, functional deterioration progresses similarly from early to late elderly stages of ageing, yet older males demonstrate superior muscle size and strength, and better motor control and functional performance. These findings add to the paucity of data in older females and emphasize the necessity for early interventions in this demographic to avert functional decline to reduce the ageing health–sex paradox.

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

## Additional information

### Data availability statement

The datasets generated and analysed during the current study are available from the corresponding author upon reasonable request.

## Competing interests

None.

## Author contributions

Y.G., J.P., P.J.A., B.E.P. and M.P. contributed to the conception and design of the work. Y.G., E.J.J., T.F.S., A.A. and N.G. acquired the data. Y.G., E.J.J., T.F.S. and A.A. analysed the data. Y.G. and M.P. drafted the manuscript and prepared the figures. Y.G. and M.P. contributed to the interpretation of the results. All authors contributed to the revision of the manuscript. All authors have approved the final version of the submitted manuscript for publication and are accountable for all aspects of the work. All persons designated as authors qualify for authorship, and all those who qualify for authorship are listed.

## Funding

This work was supported by the Medical Research Council (grant number MR/P021220/1) as part of the MRC-Versus Arthritis Centre for Musculoskeletal Ageing Research awarded to the Universities of Nottingham and Birmingham, and by the NIHR Nottingham Biomedical Research Centre.

## Acknowledgements

We thank all of the participants for their enthusiastic involvement in this study.

## Keywords

ageing, motor unit, neuromuscular function, sex differences, vastus lateralis

## Supporting information

Additional supporting information can be found online in the Supporting Information section at the end of the HTML view of the article. Supporting information files available:

**Peer Review History**

