## [Peer Review History · The Journal of Physiology]

Sex disparities of human neuromuscular decline in older humans

Yuxiao Guo, Eleanor J Jones, Thomas F Smart, Abdulmajeed Altheyab, Nishadi Gamage, Dan Stashuk, Jessica Piasecki, Bethan E. Phillips, Philip J Atherton, and Mathew Piasecki

DOI: 10.1113/JP285653

Corresponding author(s): Mathew Piasecki (mathew.piasecki@nottingham.ac.uk)

The following individual(s) involved in review of this submission have agreed to reveal their identity: J. Greig Inglis (Referee #2)

Review Timeline:

Submission Date:	08-Sep-2023
Editorial Decision:	19-Dec-2023
Revision Received:	12-Apr-2024
Editorial Decision:	29-Apr-2024
Revision Received:	03-May-2024
Accepted:	07-May-2024

Senior Editor: Richard Carson

Reviewing Editor: Madeleine Lowery

Transaction Report:

Dear Mathew,

Re: JP-RP-2023-285653 "Sex disparities in age-related neuromuscular decline: unveiling female susceptibility from early to late elderly" by Yuxiao Guo, Eleanor J Jones, Thomas F Smart, Abdulmajeed Altheyab, Nishadi Gamage, Dan Stashuk, Jessica Piasecki, Bethan E. Phillips, Philip J Atherton, and Mathew Piasecki

Thank you for submitting your manuscript to The Journal of Physiology. It has been assessed by a Reviewing Editor and by 2 expert referees and we are pleased to tell you that it is potentially acceptable for publication following satisfactory major revision.

Please address all the points raised and incorporate all requested revisions or explain in your Response to Referees why a change has not been made. We hope you will find the comments helpful and that you will be able to return your revised manuscript within 2 months. If you require longer than this, please contact journal staff: jp@physoc.org. Please note that this letter does not constitute a guarantee for acceptance of your revised manuscript.

LANGUAGE EDITING AND SUPPORT FOR PUBLICATION: If you would like help with English language editing, or other article preparation support, Wiley Editing Services offers expert help, including English Language Editing, as well as translation, manuscript formatting, and figure formatting at www.wileyauthors.com/eoo/preparation. You can also find resources for Preparing Your Article for general guidance about writing and preparing your manuscript at www.wileyauthors.com/eoo/prepresources.

REVISION CHECKLIST:

Please upload two versions of your manuscript text: one with all relevant changes highlighted and one clean version with no changes tracked. The manuscript file should include all tables and figure legends, but each figure/graph should be uploaded as separate, high-resolution files. The journal is now integrated with Wiley's Image Checking service. For further details, see: <https://www.wiley.com/en-us/network/publishing/research-publishing/trending-stories/upholding-image-integrity-wileys->

image-screening-service

We look forward to receiving your revised submission.

Best wishes,

Richard Carson
Senior Editor
The Journal of Physiology

REQUIRED ITEMS

- Author photo and profile. First or joint first authors are asked to provide a short biography (no more than 100 words for one author or 150 words in total for joint first authors) and a portrait photograph. These should be uploaded and clearly labelled together in a Word document with the revised version of the manuscript. See Information for Authors for further details.

- You must start the Methods section with a paragraph headed Ethical Approval. If experiments were conducted on humans, confirmation that informed consent was obtained, preferably in writing, that the studies conformed to the standards set by the latest revision of the Declaration of Helsinki and that the procedures were approved by a properly constituted ethics committee, which should be named, must be included in the article file. If the research study was registered (clause 35 of the Declaration of Helsinki), the registration database should be indicated, otherwise the lack of registration should be noted as an exception (e.g. The study conformed to the standards set by the Declaration of Helsinki, except for registration in a database). For further information see: <https://physoc.onlinelibrary.wiley.com/hub/human-experiments>.

- Please upload separate high-quality figure files via the submission form.

- Please include an Abstract Figure file, as well as the Figure Legend text within the main article file. The Abstract Figure is a piece of artwork designed to give readers an immediate understanding of the research and should summarise the main conclusions. If possible, the image should be easily 'readable' from left to right or top to bottom. It should show the physiological relevance of the manuscript so readers can assess the importance and content of its findings. Abstract Figures should not merely recapitulate other figures in the manuscript. Please try to keep the diagram as simple as possible and without superfluous information that may distract from the main conclusion(s). Abstract Figures must be provided by authors no later than the revised manuscript stage and should be uploaded as a separate file during online submission labelled as File Type 'Abstract Figure'. Please also ensure that you include the figure legend in the main article file. All Abstract Figures should be created using BioRender. Authors should use The Journal's premium BioRender account to export high-resolution images. Details on how to use and access the premium account are included as part of this email.

EDITOR COMMENTS

Reviewing Editor:

The reviewers are divided on the potential influence of the manuscript. Reviewer 1 has particular concerns regarding a lack of clarity surrounding the mechanisms for the differences reported and the relatively small sample size for the large spread in age, though the study addresses an understudied area. Before the manuscript could be considered for publication it would be important to address the points raised by Reviewer 1 and to emphasize how the manuscript illustrates new physiological principles or mechanisms.

REFeree COMMENTS

Referee #1:

This study aimed to investigate sex differences in age-related neuromuscular decline. Sex differences in motor unit properties were under investigated but more recently several articles have been published on the topic (e.g. 10.1111/apha.13803; 10.1152/jn.00043.2023; 10.1111/apha.14024; 10.1113/JP280679; 10.3389/fmed.2023.1185479).

My main concern with this article is that the study is deficient in terms of data richness and depth. Differences in physical performance have been largely investigated previously and the few ones observed in iEMG parameters appear of small magnitude, questioning their relevance. Moreover, the Authors did not try to explain their iEMG findings experimentally but limited their discussion to speculations. For instance, the authors did not test sex hormones concentration and other circulating and muscle biomarkers related to neurodegeneration, which could have played a role in the observed sex differences. Another factor that could be considered is about sex differences in neuromodulation as linked to persistent inward currents (PICs). Indeed, a recent paper (<https://doi.org/10.1152/jn.00043.2023>) showed that estimates of PICs were larger in females than in males. Yes, the article focused on young individuals but I believe that the authors of the present paper should consider and discuss this potential factor to explain the observed sex differences in MU discharge rate.

Overall, the manuscript is adequately written but it lacks of attempts at identifying the mechanisms underlying the main observed results. There are also a number of methodological issues that must be better clarified.

Other general comments:

- The sample size seems limited considering the large age range of the participants (56-81 years). In the title you refer to early and late elderly. Considering the small sample size and more important the fact you included middle age individuals in the 2 groups this is misleading the reader. To test the effects of aging on neuromuscular function in both sexes, the sample is also inadequate in terms of age distribution. In particular, age appears very different in the 2 groups, with the male group including 5 subjects under the age of 60 while the female group starts from 61. Even if the statistical analysis used age as a covariate this problem greatly limits the conclusions reached regarding "unveiling female susceptibility". The title should therefore be rewritten.
- The authors did not adequately address the elevated motor unit firing rate in females in the discussion section. This finding may appear counterintuitive, given the lower muscle strength in females.

Specific comments:

L37-38: Several articles showed that males and females have a similar neuromuscular decline. Probably females are frailer in ageing just because of their lower baseline levels of physical performance. The Authors are invited to rephrase this sentence.

L40: This is just a speculation, the Authors are invited to add "potentially" to the sentence. L55: Explain the rationale for the age range chosen.

L164: Explain the rationale for the short duration of the balance task. Longer durations (at least 25s) are generally recommended (e.g. <http://dx.doi.org/10.1016/j.gaitpost.2012.07.009>).

L203: The methods used for the steadiness assessment (CoV) are not fully explained, could you please expand? In particular, which kind of visual feedback was provided during the test? This information is important since your data are quite dispersed (i.e. Fig.1 E; outliers?) and quantitatively large (at least in comparison with many previously published papers) especially in the female group. Another point of discussion here is related to the fact that in the present study steadiness is more affected at the higher intensity of contraction whilst in general CoV is greatest at the lowest intensities and decreases in an exponential fashion (Hamilton et al., The scaling of motor noise with muscle strength and motor unit number in humans. *Exp Brain Res.* 2004;157; among the others).

L213: Clarify whether the needle was inserted in the muscle perpendicularly or diagonally.

The discussion should be reconsidered following the above mentioned points.

Referee #2:

The manuscript "Sex disparities in age-related neuromuscular decline: unveiling female susceptibility from early to late elderly" studies sex differences in neuromuscular decline in an elderly population. The study had 38 participants (21 male and 17 female) between the ages of 55-85. The participants performed a variety of tasks from balance, grip strength, TUG and knee extensor torque while having the iEMG monitored with an intramuscular electrode during isometric force output from the VL. This was the first study to test this population and compare the motor unit behaviour between the sexes. The results revealed older females had smaller CSA of the VL, lower knee extension torque, greater force and MUDR variability as well as higher MUDR. This led to the conclusion that females had greater deterioration compared to the male counterparts leading to the suggestion that interventions addressing these shortcomings may be required earlier in older females.

In General:

First the authors should be commended for undertaking this task in a greatly understudied area of research to help bridge the current knowledge gap in this area or study. The paper is extremely well designed and written which is to be expected from this lab group, based on previous work. I feel the paper has novel findings which are important to understand the differences between sexes throughout the ageing process. Below you will find only a few moderate and minor comments that could be used to improve the message and impact of the manuscript.

Moderate Questions:

1. Perhaps an explanation as to why both 10 and 25% MVC were chosen? It has been shown previously that lower force outputs show greater variability in force. Is it possible at these outputs you may have missed some of the larger differences between the sexes? (see Jakobi et al., review in APNM 2018)
2. Previous research on younger males and females in various muscles (ie TA and BB) at around 20%MVC have shown that females have a greater MUDR, number of motor units and variability in both force and MUDR (Inglis and Gabriel, APNM, 2020 and 2021; Harwood et al., Acta Phys, 2014, as well as your previous paper). Is it possible that these differences that you found are strictly sex differences at this level of force output which may simply continue with age? It is interesting to see the older end of the group with larger differences compared to the younger end which you did highlight. Perhaps a little more discussion regarding this at both different %MVC and in different muscle groups could bolster the message of the paper?
3. Page 17, lines 402-405: This is a great point, but is it also possible that these could be a result of sex differences in joint stability (i.e. laxity) which would inevitably increase with age? It may also be possible that the loss of estrogen may reverse the lower stiffness in the viscoelastic properties of females with age?
4. Page 18, lines 431-435: this has also been shown in other muscle groups, see references above.

Minor concerns:

1. Page 4, line 123: Remove 'the' before 'menopause'
2. Page 4, line 124: remove 'on' before 'physical function'
3. Page 5, line 134: add 'the' before 'VL'
4. Page 10, line 275: perhaps change 'non' to 'not'
5. Page 13, lines 307-311 and Page 19, line 457: change 'Mus' to 'MUs'
6. Page 13, line 320: change 'contraction' to 'contractions'
7. Throughout: at time 'MU FR' is used and others 'MUFR' please choose one

END OF COMMENTS

Sex disparities in age-related neuromuscular decline: unveiling female susceptibility from early to late elderly

The manuscript "Sex disparities in age-related neuromuscular decline: unveiling female susceptibility from early to late elderly" studies sex differences in neuromuscular decline in an elderly population. The study had 38 participants (21 male and 17 female) between the ages of 55-85. The participants performed a variety of tasks from balance, grip strength, TUG and knee extensor torque while having the iEMG monitored with an intramuscular electrode during isometric force output from the VL. This was the first study to test this population and compare the motor unit behaviour between the sexes. The results revealed older females had smaller CSA of the VL, lower knee extension torque, greater force and MUDR variability as well as higher MUDR. This led to the conclusion that females had greater deterioration compared to the male counterparts leading to the suggestion that interventions addressing these shortcomings may be required earlier in older females.

In General:

First the authors should be commended for undertaking this task in a greatly understudied area of research to help bridge the current knowledge gap in this area or study. The paper is extremely well designed and written which is to be expected from this lab group, based on previous work. I feel the paper has novel findings which are important to understand the differences between sexes throughout the ageing process. Below you will find only a few moderate and minor comments that could be used to improve the message and impact of the manuscript.

Moderate Questions:

1. Perhaps an explanation as to why both 10 and 25% MVC were chosen? It has been shown previously that lower force outputs show greater variability in force. Is it possible at these outputs you may have missed some of the larger differences between the sexes? (see Jakobi et al., review in APNM 2018)
2. Previous research on younger males and females in various muscles (ie TA and BB) at around 20%MVC have shown that females have a greater MUDR, number of motor units and variability in both force and MUDR (Inglis and Gabriel, APNM, 2020 and 2021; Harwood et al., Acta Phys, 2014, as well as your previous paper). Is it possible that these differences that you found are strictly sex differences at this level of force output which may simply continue with age? It is interesting to see the older end of the group with larger differences compared to the younger end which you did highlight. Perhaps a little more discussion regarding this at both different %MVC and in different muscle groups could bolster the message of the paper?
3. Page 17, lines 402-405: This is a great point, but is it also possible that these could be a result of sex differences in joint stability (i.e. laxity) which would inevitably increase with age? It may also be possible that the loss of estrogen may reverse the lower stiffness in the viscoelastic properties of females with age?
4. Page 18, lines 431-435: this has also been shown in other muscle groups, see references above.

Minor concerns:

Sex disparities in age-related neuromuscular decline: unveiling female susceptibility from early to late elderly

1. Page 4, line 123: Remove 'the' before 'menopause'
2. Page 4, line 124: remove 'on' before 'physical function'
3. Page 5, line 134: add 'the' before 'VL'
4. Page 10, line 275: perhaps change 'non' to 'not'
5. Page 13, lines 307-311 and Page 19, line 457: change 'Mus" to 'MUs'
6. Page 13, line 320: change 'contraction' to 'contractions'
7. Throughout: at time 'MU FR' is used and others 'MUFR' please choose one

Manuscript Number: JP-RP-2023-285653

Title: Sex disparities of human neuromuscular decline from early to late elderly

EDITOR COMMENTS

Reviewing Editor:

The reviewers are divided on the potential influence of the manuscript. Reviewer 1 has particular concerns regarding a lack of clarity surrounding the mechanisms for the differences reported and the relatively small sample size for the large spread in age, though the study addresses an understudied area. Before the manuscript could be considered for publication it would be important to address the points raised by Reviewer 1 and to emphasize how the manuscript illustrates new physiological principles or mechanisms.

We are grateful for the thorough review process and for the constructive criticism provided. The reviewer comments were substantial and in response, we have performed further experiments and analyses. To summarise the larger modifications, in response to one of the larger limitations highlighted by reviewer 1, we have removed data from five male participants under the age of 60yrs to improve age-matching across sexes. We have included seventeen further participants to increase the total N from 38 to 50 people (inclusive of the removal of all males <60yrs). This updated dataset now comprises 50 older adults (26 males (61-83yrs) and 24 females (60-79yrs)), and to our knowledge, is the largest sex and age-based examination of individual MU characteristics. As such, we now include a larger sample size within a smaller age-range. The updated data analysis has had little effect on the majority of conclusions, however perhaps the most notable change is regression models indicate the sex difference in MUFR variability is now not statistically significant (previous $p=0.031$, updated $p=0.085$).

Responses to individual comments are shown below in red, and updates to the original manuscript (attached) are also shown in red.

REFEREE COMMENTS

Referee #1:

This study aimed to investigate sex differences in age-related neuromuscular decline. Sex differences in motor unit properties were under investigated but more recently several articles have been published on the topic (e.g. 10.1111/apha.13803; 10.1152/jn.00043.2023; 10.1111/apha.14024; 10.1113/JP280679; 10.3389/fmed.2023.1185479).

We agree this is a rapidly expanding research area and we hope the inclusion of age in the current manuscript adds to this field.

My main concern with this article is that the study is deficient in terms of data richness and depth. Differences in physical performance have been largely investigated previously and the few ones observed in iEMG parameters appear of small magnitude, questioning their

relevance. Moreover, the Authors did not try to explain their iEMG findings experimentally but limited their discussion to speculations. For instance, the authors did not test sex hormones concentration and other circulating and muscle biomarkers related to neurodegeneration, which could have played a role in the observed sex differences. Another factor that could be considered is about sex differences in neuromodulation as linked to persistent inward currents (PICs). Indeed, a recent paper (<https://doi.org/10.1152/jn.00043.2023>) showed that estimates of PICs were larger in females than in males. Yes, the article focused on young individuals but I believe that the authors of the present paper should consider and discuss this potential factor to explain the observed sex differences in MU discharge rate.

Overall, the manuscript is adequately written but it lacks of attempts at identifying the mechanisms underlying the main observed results. There are also a number of methodological issues that must be better clarified.

Thank you for your insight, we are grateful for the opportunity to address your concerns and improve the quality of our study. Upon further reading we agreed we have put too much emphasis on the less novel aspects of these data which detracts from the importance of the study. We feel the expanded N, updated analysis and text amendments provide for a much-improved manuscript.

We understood your concern regarding the small sample size in our study to explore sex disparities from early to late elderly individuals. To address these issues, we have revised our approach by excluding five older males under the age of 60 and including 17 additional older male and female participants, resulting in a total of 26 older males and 24 older females.

Following our previous findings in a younger cohort, this study was aiming to further explore the sex difference in neuromuscular function with ageing and how increasing age would influence it. We believe our study contributes to the limited body of knowledge concerning the aging female population in particular.

We agree hormone concentrations may have been useful but remain doubtful clear correlations could be made with MU properties. We also believe there are currently no reliable circulating biomarkers of MU function. However, this is a rapidly expanding field and exploration of this is incorporated into future plans, alongside the determinants of higher MUF in older females, including estimates of persistent inward currents (i.e. sustained firing), and the effects of hormone replacement therapy (HRT). Moreover, all participants in this study were not receiving any form of HRT and were all post-menopausal, which minimises probable effects of variable hormone concentrations in females.

As we agree these additional measures may have strengthened aspects of the study, we have highlighted this within the manuscript.

Line 472 now reads:

“Secondly, we did not measure the circulating levels of sex hormones nor other circulating biomarkers related to neural plasticity.”

With the inclusion of additional data, we still observed a larger MUFR in older females, and we have amended this section to properly discuss this sex disparities, which now reads:

Line 372:

“Females exhibited significant higher MUFR than males at normalised contraction levels, and this difference across sex was over two-fold larger than the difference from low to mid-level contractions (beta: *sex* 0.81 v *level* 0.33). This higher MUFR in females is consistent with numerous studies of young previously observed in both upper and lower extremities (Harwood *et al.*, 2014; Peng *et al.*, 2018; Inglis & Gabriel, 2021; Taylor *et al.*, 2022; Guo *et al.*, 2022a; Nishikawa *et al.*, 2024). We previously reported in young VL a higher MUFR in females, with a smaller MUP and similar MU number estimates. Here we suggested females compensate for smaller MUs by discharging at higher rates (Guo *et al.*, 2022). However, in the current study of older people, there was no significant difference in MUP area between sexes, which may be a result of MU remodelling (i.e. expansion) occurring in both sexes (Jones *et al.*, 2022). Therefore, the same argument of compensating for smaller MUs lacks conviction in old.

Myosteatosis (fatty infiltration of muscle) increases in older age and is associated with impaired neuromuscular function (Aoi *et al.*, 2020). Females typically exhibit higher levels of myosteatosis compared to males, irrespective of variations in BMI or overall body fat levels (Goodpaster *et al.*, 2001; Ryan *et al.*, 2011; Therkelsen *et al.*, 2013; Xiao *et al.*, 2018). More specifically, as intramuscular adipose tissue increases, even when adjusted for muscle CSA, muscle quality declines (Biltz *et al.*, 2020). Therefore, the elevated MUFR observed in older females may represent a compensatory mechanism for lower muscle quality.

Differences in MUFR may stem from the organisation of central and peripheral inputs into the motoneuron pool and/or differences in the intrinsic characteristics of MUs. Persistent inward currents (PICs) play a critical role in initiating sustained firing of motoneurons under conditions of descending monoaminergic drive (Heckman *et al.*, 2008). Estimates of the PIC contribution to MU firing made via the deltaF technique (Gorassini *et al.*, 2002) are consistently lower in old males compared to young (Hassan *et al.*, 2021; Orssatto *et al.*, 2022, 2023; Guo *et al.*, 2024) and are greater in young females compared to young males (Jenz *et al.*, 2023). Given the sex differences of MUFR widely observed in young are also apparent in old, it is probable that estimates of PIC contribution to MU firing also maintain a sex difference in older age. Although the deltaF technique was not used here, these data suggest minimal effects of the menopause on MUFR at these low and mid-level contraction intensities.

This may not be true of all contraction levels and/or muscles; in lower limbs- specifically the vastus medialis, TA and VL, females exhibited higher MUFR during submaximal contractions, ranging from 10% to 75% of MVC (Peng *et al.*, 2018; Kowalski & Anita D., 2020; Inglis & Gabriel, 2020; Taylor *et al.*, 2022; Guo *et al.*, 2022). However, at 100% MVC, females showed lower MUFR than males (Inglis & Gabriel, 2020). With force increases, we found that recruitment strategies for early and late elderly males and females did not differ, also reported for young cohorts (Guo *et al.*, 2022). Put simply, when moving from a low to a mid-level contraction, the change in MU rate coding and recruitment did not differ between sexes, as indicated by higher MUFR and larger MUP area in males and females.”

Other general comments:

- The sample size seems limited considering the large age range of the participants (56-81 years). In the title you refer to early and late elderly. Considering the small sample size and more important the fact you included middle age individuals in the 2 groups this is misleading the reader. To test the effects of aging on neuromuscular function in both sexes, the sample is also inadequate in terms of age distribution. In particular, age appears very different in the 2 groups, with the male group including 5 subjects under the age of 60 while the female group starts from 61. Even if the statistical analysis used age as a covariate this problem greatly limits the conclusions reached regarding "unveiling female susceptibility". The title should therefore be rewritten.

This is a very good point and we have made efforts to address it by increasing the sample size and improving the age-matching across sexes.

We have revised our approach by removing five males under the age of 60 and including 17 additional older male and female participants, resulting in a total of 26 older males and 24 older females in the study. As advised, we have also updated the title of the manuscript to: "Sex disparities of human neuromuscular decline from early to late elderly".

- The authors did not adequately address the elevated motor unit firing rate in females in the discussion section. This finding may appear counterintuitive, given the lower muscle strength in females.

Thank you for highlighting this. We have amended the corresponding paragraph to better address this difference. However, we remain doubtful MUFR at submaximal levels can directly explain maximal strength differences. This section now reads:

"Females exhibited significant higher MUFR than males at normalised contraction levels, and this difference across sex was over two-fold larger than the difference from low to mid-level contractions (beta: sex 0.81 v level 0.33). This higher MUFR in females is consistent with numerous studies of young previously observed in both upper and lower extremities (Harwood et al., 2014; Peng et al., 2018; Inglis & Gabriel, 2021; Taylor et al., 2022; Guo et al., 2022a; Nishikawa et al., 2024). We previously reported in young VL a higher MUFR in females, with a smaller MUP and similar MU number estimates. Here we suggested females compensate for smaller MUs by discharging at higher rates (Guo et al., 2022). However, in the current study of older people, there was no significant difference in MUP area between sexes, which may be a result of MU remodelling (i.e. expansion) occurring in both sexes (Jones et al., 2022). Therefore, the same argument of compensating for smaller MUs lacks conviction in old.

Myosteatosis (fatty infiltration of muscle) increases in older age and is associated with impaired neuromuscular function (Aoi et al., 2020). Females typically exhibit higher levels of myosteatosis compared to males, irrespective of variations in BMI or overall body fat levels (Goodpaster et al., 2001; Ryan et al., 2011; Therkelsen et al., 2013; Xiao et al., 2018). More specifically, as intramuscular adipose tissue increases, even when adjusted for muscle CSA,

muscle quality declines (Biltz *et al.*, 2020). Therefore, the elevated MUFR observed in older females may represent a compensatory mechanism for lower muscle quality.

Differences in MUFR may stem from the organisation of central and peripheral inputs into the motoneuron pool and/or differences in the intrinsic characteristics of MUs. Persistent inward currents (PICs) play a critical role in initiating sustained firing of motoneurons under conditions of descending monoaminergic drive (Heckman *et al.*, 2008). Estimates of the PIC contribution to MU firing made via the deltaF technique (Gorassini *et al.*, 2002) are consistently lower in old males compared to young (Hassan *et al.*, 2021; Orsatto *et al.*, 2022, 2023; Guo *et al.*, 2024) and are greater in young females compared to young males (Jenz *et al.*, 2023). Given the sex differences of MUFR widely observed in young are also apparent in old, it is probable that estimates of PIC contribution to MU firing also maintain a sex difference in older age. Although the deltaF technique was not used here, these data suggest minimal effects of the menopause on MUFR at these low and mid-level contraction intensities.

This may not be true of all contraction levels and/or muscles; in lower limbs- specifically the vastus medialis, TA and VL, females exhibited higher MUFR during submaximal contractions, ranging from 10% to 75% of MVC (Peng *et al.*, 2018; Kowalski & Anita D., 2020; Inglis & Gabriel, 2020; Taylor *et al.*, 2022; Guo *et al.*, 2022). However, at 100% MVC, females showed lower MUFR than males (Inglis & Gabriel, 2020). With force increases, we found that recruitment strategies for early and late elderly males and females did not differ, also reported for young cohorts (Guo *et al.*, 2022). Put simply, when moving from a low to a mid-level contraction, the change in MU rate coding and recruitment did not differ between sexes, as indicated by higher MUFR and larger MUP area in males and females.”

Specific comments:

L37-38: Several articles showed that males and females have a similar neuromuscular decline. Probably females are frailer in ageing just because of their lower baseline levels of physical performance. The Authors are invited to rephrase this sentence.

Thank you for highlighting this. We agree this is an oversimplification and lacks specificity. We have amended this sentence, which now reads:

Line 31:

“Females typically live longer than males but paradoxically, spend a greater number of later years in poorer health.”

L40: This is just a speculation, the Authors are invited to add "potentially" to the sentence.

Agreed. We have amended this sentence, which now reads:

Line 32:

“The neuromuscular system is a critical component of the progression to frailty, and motor unit (MU) characteristics differ by sex in healthy young people and may adapt to ageing in a sex-specific manner due to divergent hormonal profiles.”

L55: Explain the rationale for the age range chosen.

We have amended all of the abstract but feel this is not the space to outline the rationale for investigating older humans. However, we have further addressed this at several points throughout the manuscript. The abstract conclusion now reads:

“Higher VL MUFR at normalised contraction levels previously observed in young are also apparent in old, with no sex-based difference of estimates of MU structure or NMJ transmission instability. From early to late elderly, the deterioration of neuromuscular function and MU characteristics did not differ between sexes, yet function was consistently greater in males. These parallel trajectories underscore the lower initial level for older females and may offer insights into identifying critical intervention periods.”

Line 128 of the introduction now reads:

“Furthermore, age comparisons are frequently conducted through cross-sectional studies comparing young and old groups, wherein the older group typically encompasses an age range where physiological deterioration of MU function may be anticipated (Hirono *et al.*, 2024).”

L164: Explain the rationale for the short duration of the balance task. Longer durations (at least 25s) are generally recommended(e.g. <http://dx.doi.org/10.1016/j.gaitpost.2012.07.009>).

We understand the importance of using longer duration (at least 25s) to capture more robust data, however this is unrealistic for many older individuals and the outcomes of the cited study may be less applicable to these and other clinical cohorts. We selected 10s as all older individuals were able to perform this. Additionally, 10s balance has been reported to be independently associated with all-cause mortality in middle-aged and older adults (Araujo *et al.*, 2022).

L203: The methods used for the steadiness assessment (CoV) are not fully explained, could you please expand? In particular, which kind of visual feedback was provided during the test? This information is important since your data are quite dispersed (i.e. Fig.1 E; outliers?) and quantitatively large (at least in comparison with many previously published papers) especially in the female group. Another point of discussion here is related to the fact that in the present study steadiness is more affected at the higher intensity of contraction whilst in general CoV is greatest at the lowest intensities and decreases in an exponential fashion

(Hamilton et al., The scaling of motor noise with muscle strength and motor unit number in humans. Exp Brain Res. 2004;157; among the others).

Force steadiness was recorded during voluntary isometric contractions normalised to MVC. For each contraction intensity, a single target line was visible on a screen and participants were instructed to follow as closely as possible. Contractions were held for 12s followed by a 20s rest. For analysis purpose, the first two passes of the target line (<1s) were excluded from calculation to avoid corrective actions when reaching the target line. We have added further information in the methods section.

Line 206 now reads:

“To quantify force steadiness, a single target line at a normalised contraction intensity was displayed on a screen and the participant was instructed to follow as closely as possible. The coefficient of variation of the force (CoV) was calculated = $(SD/Mean)*100$. When calculating CoV, in an attempt to reduce corrective actions, the first two passes of the target (<1s) were excluded from the analysis.”

In relation to the second point, force steadiness (CoV) was lower at 25% when compared to 10% for all participants. We may have misunderstood, but this appears to agree with the reviewer’s point, that FS reduces with increasing contraction intensity. We also note this consistent finding in all our studies, reflected by the higher mean values at higher contractions typically resulting in lower CoV. With the addition of further data and updated statistical analyses, we still observe significant improvements in force steadiness transitioning from low to mid-level contractions ($p<0.009$). This is shown in Table 2.

L213: Clarify whether the needle was inserted in the muscle perpendicularly or diagonally.

We have clarified that the needle was perpendicularly inserted into the muscle belly of VL of the right leg.

The discussion should be reconsidered following the above mentioned points.

Agreed. We have amended the majority of the discussion to reflect these and other points.

Referee #2:

The manuscript "Sex disparities in age-related neuromuscular decline: unveiling female susceptibility from early to late elderly" studies sex differences in neuromuscular decline in an elderly population. The study had 38 participants (21 male and 17 female) between the ages of 55-85. The participants performed a variety of tasks from balance, grip strength, TUG and knee extensor torque while having the iEMG monitored with an intramuscular electrode during isometric force output from the VL. This was the first study to test this population and compare the motor unit behaviour between the sexes. The results revealed older females had smaller CSA of the VL, lower knee extension torque, greater force and MUDR

variability as well as higher MUDR. This led to the conclusion that females had greater deterioration compared to the male counterparts leading to the suggestion that interventions addressing these short comings may be required earlier in older females.

In General:

First the authors should be commended for undertaking this task in a greatly understudied area of research to help bridge the current knowledge gap in this area or study. The paper is extremely well designed and written which is to be expected from this lab group, based on previous work. I feel the paper has novel findings which are important to understand the differences between sexes throughout the ageing process. Below you will find only a few moderate and minor comments that could be used to improve the message and impact of the manuscript.

Thank you for your review. We have addressed individual comments below.

Moderate Questions:

1. Perhaps an explanation as to why both 10 and 25% MVC were chosen? It has been shown previously that lower force outputs show greater variability in force. Is it possible at these outputs you may have missed some of the larger differences between the sexes? (see Jakobi et al., review in APNM 2018)

Thank you for highlighting this. 10% and 25% MVC were chosen as this range is representative of activities of daily living, from walking to climbing stairs (Tikkanen *et al.*, 2013), crucial for ageing individuals. We also intended to explore the difference between the two contraction levels as a function of sex, having noted recruitment strategies in younger populations do not differ between males and females (Guo *et al.*, 2022). Finally, we also note that 26-gauge intramuscular electrodes are tolerable within this range and also provide a high MU yield. Although higher contraction intensities are limited with this method, we emphasise that in older VL, the large amount of subcutaneous tissue around the thigh (particularly females) renders surface-based measures of MU activity unreliable.

We have addressed these points in several areas of the manuscript.

Line 225:

“These contraction levels were chosen as this range is generally representative of activities of daily living such as walking and climbing stairs (Tikkanen *et al.*, 2013) and is known to be both tolerable and to provide a high MU yield (Guo *et al.*, 2022).”

2. Previous research on younger males and females in various muscles (ie TA and BB) at around 20%MVC have shown that females have a greater MUDR, number of motor units and variability in both force and MUDR (Inglis and Gabriel, APNM, 2020 and 2021; Harwood et al., Acta Phys, 2014, as well as your previous paper). Is it possible that these differences that you found are strictly sex differences at this level of force output which may simply continue

with age? It is interesting to see the older end of the group with larger differences compared to the younger end which you did highlight. Perhaps a little more discussion regarding this at both different %MVC and in different muscle groups could bolster the message of the paper?

Thank you for highlighting this. We have added further discussion around the possible influence of contraction intensity.

Line 404:

“This may not be true of all contraction levels and/or muscles; in lower limbs- specifically the vastus medialis, TA and VL, females exhibited higher MUFR during submaximal contractions, ranging from 10% to 75% of MVC (Peng *et al.*, 2018; Kowalski & Anita D., 2020; Inglis & Gabriel, 2020; Taylor *et al.*, 2022; Guo *et al.*, 2022). However, at 100% MVC, females showed lower MUFR than males (Inglis & Gabriel, 2020). With force increases, we found that recruitment strategies for early and late elderly males and females did not differ, also reported for young cohorts (Guo *et al.*, 2022). Put simply, when moving from a low to a mid-level contraction, the change in MU rate coding and recruitment did not differ between sexes, as indicated by higher MUFR and larger MUP area in males and females.”

3. Page 17, lines 402-405: This is a great point, but is it also possible that these could be a result of sex differences in joint stability (i.e. laxity) which would inevitably increase with age? It may also be possible that the loss of estrogen may reverse the lower stiffness in the viscoelastic properties of females with age?

Thank you for highlighting this. We have included further discussion on these points. This section now reads:

Line 426:

“Age-related changes in mechanical properties of joints play a crucial role in regulating force and control ability. Greater muscle tone and reduced elasticity in upper limbs (Lee *et al.*, 2022) as well as heightened co-contraction of agonist and antagonist muscles in lower limbs (Hortobágyi & DeVita, 2006; Krishnan *et al.*, 2011), have been widely reported with age. These changes lead to elevated joint contact forces (Hodge *et al.*, 1986) and increased impact loads (Lafortune *et al.*, 1996). Moreover, a decline in collagen content with age may contribute to increased tendon and ligament laxity, potentially exacerbating muscle co-contraction patterns (Rudolph *et al.*, 2007).”

4. Page 18, lines 431-435: this has also been shown in other muscle groups, see references above.

We have included these relevant studies reporting the similar findings in various muscle groups.

Minor concerns:

1. Page 4, line 123: Remove 'the' before 'menopause'
2. Page 4, line 124: remove 'on' before 'physical function'
3. Page 5, line 134: add 'the' before 'VL'
4. Page 10, line 275: perhaps change 'non' to 'not'
5. Page 13, lines 307-311 and Page 19, line 457: change 'Mus' to 'MUs'
6. Page 13, line 320: change 'contraction' to 'contractions'
7. Throughout: at time 'MU FR' is used and others 'MUFR' please choose one

Thank you for highlighting these errors, all of which have now been corrected.

Thank you for your patience with our progress.

Reference

- Aoi W, Rittweger J, Feng H-Z, Addison O, Correa-de-Araujo R, Miljkovic I, Goodpaster B, Bergman B, Clark R, Elena J, Esser K, Ferrucci L, Harris-Love M, Kritchevsky S, Lorbergs A, Shepherd J, Shulman G & Rosen C (2020). Myosteatorsis in the Context of Skeletal Muscle Function Deficit: An Interdisciplinary Workshop at the National Institute on Aging. *Front Physiol* **11**, 1–18.
- Araujo CG, de Souza e Silva C, Laukkanen J, Singh M, Kunutsor S, Myers J, Franca J & Castro C (2022). Successful 10-second one-legged stance performance predicts survival in middle-aged and older individuals. *Br J Sports Med* **56**, bjsports-2021.
- Biltz N, Collins K, Shen K, Schwartz K, Harris C & Meyer G (2020). Infiltration of intramuscular adipose tissue impairs skeletal muscle contraction. *J Physiol*; DOI: 10.1113/jp279595.
- Goodpaster B, Carlson C, Visser M, Kelley D, Scherzinger A, Harris T, Stamm E & Newman A (2001). Attenuation of skeletal muscle and strength in the elderly: The Health ABC Study. *J Appl Physiol* **90**, 2157.
- Gorassini M, Yang JF, Siu M & Bennett DJ (2002). Intrinsic Activation of Human Motoneurons: Possible Contribution to Motor Unit Excitation. *J Neurophysiol* **87**, 1850–1858.
- Guo Y, Jones EJ, Inns TB, Ely IA, Stashuk DW, Wilkinson DJ, Smith K, Piasecki J, Phillips BE, Atherton PJ & Piasecki M (2022). Neuromuscular recruitment strategies of the vastus lateralis according to sex. *Acta Physiologica* **235**, e13803.

- Guo Y, Jones EJ, Škarabot J, Inns TB, Phillips BE, Atherton PJ & Piasecki M (2024). Common synaptic inputs and persistent inward currents of vastus lateralis motor units are reduced in older male adults. *Geroscience*; DOI: 10.1007/s11357-024-01063-w.
- Harwood B, Cornett KMD, Edwards DL, Brown RE & Jakobi JM (2014). The effect of tendon vibration on motor unit activity, intermuscular coherence and force steadiness in the elbow flexors of males and females. *Acta Physiol (Oxf)* **211**, 597–608.
- Hassan AS, Fajardo ME, Cummings M, McPherson LM, Negro F, Dewald JPA, Heckman CJ & Pearcey GEP (2021). Estimates of persistent inward currents are reduced in upper limb motor units of older adults. *J Physiol* **599**, 4865–4882.
- Heckman CJ, Johnson M, Mottram C & Schuster J (2008). Persistent inward currents in spinal motoneurons and their influence on human motoneuron firing patterns. *Neuroscientist* **14**, 264–275.
- Hirono T, Takeda R, Nishikawa T & Watanabe K (2024). Prediction of 1-year change in knee extension strength by neuromuscular properties in older adults. *Geroscience* **46**, 2561–2569.
- Hodge WA, Fijan RS, Carlson KL, Burgess RG, Harris WH & Mann RW (1986). Contact pressures in the human hip joint measured in vivo. *Proc Natl Acad Sci U S A* **83**, 2879–2883.
- Hortobágyi T & DeVita P (2006). Mechanisms Responsible for the Age-Associated Increase in Coactivation of Antagonist Muscles. *Exerc Sport Sci Rev.*
- Inglis JG & Gabriel DA (2020). Sex differences in motor unit discharge rates at maximal and submaximal levels of force output. *Applied Physiology, Nutrition, and Metabolism* **45**, 1197–1207.
- Inglis JG & Gabriel DA (2021). Sex differences in the modulation of the motor unit discharge rate leads to reduced force steadiness. *Applied Physiology, Nutrition, and Metabolism* **46**, 1065–1072.
- Jenz ST, Beauchamp JA, Gomes MM, Negro F, Heckman CJ & Pearcey GEP (2023). Estimates of persistent inward currents in lower limb motoneurons are larger in females than in males. *J Neurophysiol* **129**, 1322–1333.
- Jones EJ, Chiou S-Y, Atherton PJ, Phillips BE & Piasecki M (2022). Ageing and exercise-induced motor unit remodelling. *J Physiol* **600**, 1839–1849.
- Kowalski KL & Anita D. C (2020). Force Control and Motor Unit Firing Behavior Following Mental Fatigue in Young Female and Male Adults. *Front Integr Neurosci*. Available at: <https://www.frontiersin.org/articles/10.3389/fnint.2020.00015>.
- Krishnan C, Allen EJ & Williams GN (2011). Effect of knee position on quadriceps muscle force steadiness and activation strategies. *Muscle Nerve* **43**, 563–573.

- Lafortune MA, Hennig EM & Lake MJ (1996). Dominant role of interface over knee angle for cushioning impact loading and regulating initial leg stiffness. *J Biomech* **29**, 1523–1529.
- Lee M-T, Wu C-Y, Chen C-W, Cheng H-L, Chen C-C & Hsieh Y-W (2022). Age and sex differences in the biomechanical and viscoelastic properties of upper limb muscles in middle-aged and older adults: A pilot study. *J Biomech* **134**, 111002.
- Orssatto LBR, Blazeovich AJ & Trajano GS (2023). Ageing reduces persistent inward current contribution to motor neurone firing: Potential mechanisms and the role of exercise. *J Physiol* **601**, 3705–3716.
- Orssatto LBR, Fernandes GL, Blazeovich AJ & Trajano GS (2022). Facilitation–inhibition control of motor neuronal persistent inward currents in young and older adults. *J Physiol* **600**, 5101–5117.
- Peng Y-L, Tenan MS & Griffin L (2018). Hip position and sex differences in motor unit firing patterns of the vastus medialis and vastus medialis oblique in healthy individuals. *J Appl Physiol* **124**, 1438–1446.
- Rudolph KS, Schmitt LC & Lewek MD (2007). Age-Related Changes in Strength, Joint Laxity, and Walking Patterns: Are They Related to Knee Osteoarthritis? *Phys Ther* **87**, 1422–1432.
- Ryan AS, Buscemi A, Forrester L, Hafer-Macko CE & Ivey FM (2011). Atrophy and Intramuscular Fat in Specific Muscles of the Thigh: Associated Weakness and Hyperinsulinemia in Stroke Survivors. *Neurorehabil Neural Repair* **25**, 865–872.
- Taylor CA, Kopicko BH, Negro F & Thompson CK (2022). Sex differences in the detection of motor unit action potentials identified using high-density surface electromyography. *Journal of Electromyography and Kinesiology* **65**, 102675.
- Therkelsen KE, Pedley A, Speliotes EK, Massaro JM, Murabito J, Hoffmann U & Fox CS (2013). Intramuscular Fat and Associations With Metabolic Risk Factors in the Framingham Heart Study. *Arterioscler Thromb Vasc Biol* **33**, 863–870.
- Tikkanen O, Haakana P, Pesola A, Häkkinen K, Rantalainen T, Havu M, Pullinen T & Finni T (2013). Muscle Activity and Inactivity Periods during Normal Daily Life. *PLoS One* **8**, e52228.
- Xiao J, Caan BJ, Weltzien E, Cespedes Feliciano EM, Kroenke CH, Meyerhardt JA, Baracos VE, Kwan ML, Castillo AL & Prado CM (2018). Associations of pre-existing co-morbidities with skeletal muscle mass and radiodensity in patients with non-metastatic colorectal cancer. *J Cachexia Sarcopenia Muscle* **9**, 654–663.

Dear Mathew

Re: JP-RP-2024-285653R1 "Sex disparities of human neuromuscular decline from early to late elderly" by Yuxiao Guo, Eleanor J Jones, Thomas F Smart, Abdulmajeed Altheyab, Nishadi Gamage, Dan Stashuk, Jessica Piasecki, Bethan E. Phillips, Philip J Atherton, and Mathew Piasecki

Thank you for submitting your manuscript to The Journal of Physiology. It has been assessed by a Reviewing Editor and by 2 expert referees and we are pleased to tell you that it is acceptable for publication following satisfactory revision.

REVISION CHECKLIST:

Please upload two versions of your manuscript text: one with all relevant changes highlighted and one clean version with no changes tracked. The manuscript file should include all tables and figure legends, but each figure/graph should be uploaded as separate, high-resolution files. The journal is now integrated with Wiley's Image Checking service. For further details, see: <https://www.wiley.com/en-us/network/publishing/research-publishing/trending-stories/upholding-image-integrity-wileys-image-screening-service>.

- 'Potential Cover Art' for consideration as the issue's cover image
- Appropriate Supporting Information (video, audio or data set: see https://jp.msubmit.net/cgi-bin/main.plex?form_type=display_requirements#supp)

We look forward to receiving your revised submission.

Best wishes,

Richard Carson
Senior Editor
The Journal of Physiology

REQUIRED ITEMS

- You must start the Methods section with a paragraph headed Ethical Approval. If experiments were conducted on humans, confirmation that informed consent was obtained, preferably in writing, that the studies conformed to the standards set by the latest revision of the Declaration of Helsinki and that the procedures were approved by a properly constituted ethics committee, which should be named, must be included in the article file. If the research study was registered (clause 35 of the Declaration of Helsinki), the registration database should be indicated, otherwise the lack of registration should be noted as an exception (e.g. The study conformed to the standards set by the Declaration of Helsinki, except for registration in a database). For further information see: <https://physoc.onlinelibrary.wiley.com/hub/human-experiments>.

EDITOR COMMENTS

Reviewing Editor:

Both Referees have reviewed the revised manuscript and agree that it has been substantially improved. Most of their original comments have been addressed. A small number of outstanding points remain which should be considered, in particular the concern raised by Referee 1 regarding the analysis of changes associated with early to late aging.

Please include a 'clause 35' statement in the Methods (see Required Items above).

REFEREE COMMENTS

Referee #1:

The study has improved significantly. However, some points remain to be considered. Yes, it is true the sample has increased and subjects under 60 years of age have been excluded, however the sample size is still not sufficient to justify an early to late aging analysis. In this sense I would argue that the title should be limited to: "Sex disparities of human neuromuscular decline in older individuals" so removing early to late elderly.

In this regard it would be appropriate to present the MUFR data also as scatter plots of individual values as a function of age at the two intensities, this would strengthen the representation of an effect of age (early to advanced) on the parameter in the two sexes. Fig. 2 does not currently allow for such a view.

Referee #2:

The authors have done an admirable job in addressing the comments of both reviewers in my opinion. Below are a couple of very minor comments.

Page 18, line 383: 'significant' should be 'significantly'

Page 19, line 420. Although this statement is correct. In the Inglis and Gabriel paper, when the sexes were strength and physical activity matched females continued to have a greater MUDR compared to males. You may want to mention this here.

END OF COMMENTS

1st Confidential Review

12-Apr-2024

Page 18, line 383: 'significant' should be 'significantly'

Page 19, line 420. Although this statement is correct. In the Inglis and Gabriel paper, when the sexes were strength and physical activity matched females continued to have a greater MUDR compared to males. You may want to mention this here.

Manuscript Number: JP-RP-2024-285653

Title: Sex disparities of human neuromuscular decline in older humans

We are very grateful to the reviewers and editor for their prompt and thorough evaluation of our manuscript. The manuscript has been further modified in response to the comments and we trust it is now suitable for publication.

EDITOR COMMENTS

Reviewing Editor:

Both Referees have reviewed the revised manuscript and agree that it has been substantially improved. Most of their original comments have been addressed. A small number of outstanding points remain which should be considered, in particular the concern raised by Referee 1 regarding the analysis of changes associated with early to late aging.

Please include a 'clause 35' statement in the Methods (see Required Items above).

Thank you for the prompt handling and thorough assessment of our manuscript. We have addressed individual comments below. We have also included a 'clause 35' statement in methods section which now adhere to this Journal's policies.

REFEREE COMMENTS

Referee #1:

The study has improved significantly. However, some points remain to be considered. Yes, it is true the sample has increased and subjects under 60 years of age have been excluded, however the sample size is still not sufficient to justify an early to late aging analysis. In this sense I would argue that the title should be limited to: "Sex disparities of human neuromuscular decline in older individuals" so removing early to late elderly.

As advised, we have updated the title of the manuscript to

Sex disparities of human neuromuscular decline in older humans

In this regard it would be appropriate to present the MUFR data also as scatter plots of individual values as a function of age at the two intensities, this would strengthen the

representation of an effect of age (early to advanced) on the parameter in the two sexes. Fig. 2 does not currently allow for such a view.

Thank you for highlighting this. After adjusting for sex and contraction level, there was no association between age and all MU parameters (all $p > 0.05$). Hence, we placed more emphasis on differences based on sex and contraction level in this older cohort. Considering your feedback, we acknowledge that including an additional figure illustrating the impact of age would enhance the clarity of our findings. To visually represent the effect of age on MU function, we have included an additional 12 panel figure displaying all recorded MUPs and participant means. This is now Figure 2 within the manuscript and is shown below.

Figure 2. Measurement of motor unit structure and function in older males (pink) and females (purple) aged 60-83 years. All motor unit action potentials (MUPs) recorded during contractions at 10% and 25% maximal voluntary contraction (MVC) are shown as jitter points. Individual participant means are shown as scatter points (black) with males in circles and females in triangles. Grey lines indicate the beta coefficient of age from multilevel mixed effects linear regression models with an adjustment of sex.

Referee #2:

The authors have done an admirable job in addressing the comments of both reviewers in my opinion. Below are a couple of very minor comments.

Thank you for the positive overview and comments.

Page 18, line 383: 'significant' should be 'significantly'.

Thank you, we have corrected this typo.

Page 19, line 420. Although this statement is correct. In the Inglis and Gabriel paper, when the sexes were strength and physical activity matched females continued to have a greater MUDR compared to males. You may want to mention this here.

Thank you for highlighting this. We have amended this sentence, which now reads:

“Conversely, at 100% MVC, females showed lower MUFR than males. Notably, when strength and physical activity were balanced between the sexes, females consistently maintained a higher MUFR than males (Inglis & Gabriel, 2020).”

Dear Dr Piasecki,

Re: JP-RP-2024-285653R2 "Sex disparities of human neuromuscular decline in older humans" by Yuxiao Guo, Eleanor J Jones, Thomas F Smart, Abdulmajeed Altheyab, Nishadi Gamage, Dan Stashuk, Jessica Piasecki, Bethan E. Phillips, Philip J Atherton, and Mathew Piasecki

We are pleased to tell you that your paper has been accepted for publication in The Journal of Physiology.

Authors should note that it is too late at this point to offer corrections prior to proofing. Major corrections at proof stage, such as changes to figures, will be referred to the Editors for approval before they can be incorporated. Only minor changes, such as to style and consistency, should be made at proof stage. Changes that need to be made after proof stage will usually require a formal correction notice.

If you would like to receive our 'Research Roundup', a monthly newsletter highlighting the cutting-edge research published in The Physiological Society's family of journals (The Journal of Physiology, Experimental Physiology and Physiological Reports), please click this link, fill in your name and email address and select 'Research Roundup': <https://www.physoc.org/journals-and-media/membernews/>.

Yours sincerely,

Richard Carson
Senior Editor
The Journal of Physiology

P.S. - You can help your research get the attention it deserves! Check out Wiley's free Promotion Guide for best-practice recommendations for promoting your work at www.wileyauthors.com/eoo/guide. You can learn more about Wiley Editing Services which offers professional video, design, and writing services to create shareable video abstracts, infographics, conference posters, lay summaries, and research news stories for your research at www.wileyauthors.com/eoo/promotion.

IMPORTANT NOTICE ABOUT OPEN ACCESS: To assist authors whose funding agencies mandate public access to published research findings sooner than 12 months after publication, The Journal of Physiology allows authors to pay an Open Access (OA) fee to have their papers made freely available immediately on publication.

You can check if your funder or institution has a Wiley Open Access Account here: <https://authorservices.wiley.com/author-resources/Journal-Authors/licensing-and-open-access/open-access/author-compliance-tool.html>.

EDITOR COMMENTS

Reviewing Editor:

The authors have satisfactorily addressed the Referees' last remaining comments.